# ICBrainDB: An Integrated Database for Finding Associations between Genetic Factors and EEG Markers of Depressive Disorders

**DOI:** 10.3390/jpm12010053

**Published:** 2022-01-05

**Authors:** Roman Ivanov, Fedor Kazantsev, Evgeny Zavarzin, Alexandra Klimenko, Natalya Milakhina, Yury G. Matushkin, Alexander Savostyanov, Sergey Lashin

**Affiliations:** 1Institute of Cytology and Genetics SB RAS, Novosibirsk 630090, Russia; kazfdr@bionet.nsc.ru (F.K.); klimenko@bionet.nsc.ru (A.K.); tashamilka@mail.ru (N.M.); mat@bionet.nsc.ru (Y.G.M.); a-sav@mail.ru (A.S.); lashin@bionet.nsc.ru (S.L.); 2Natural Science Faculty, Novosibirsk State University, Novosibirsk 630090, Russia; zavarzinevg@gmail.com; 3Kurchatov Genomics Center, Institute of Cytology and Genetics, Siberian Branch of the Russian Academy of Science, Lavrentiev Avenue 10, Novosibirsk 630090, Russia; 4Institute of Neuroscience and Medicine, Novosibirsk 630117, Russia

**Keywords:** depression, database, EEG, SNP, questionnaires

## Abstract

In this study, we collected and systemized diverse information related to depressive and anxiety disorders as the first step on the way to investigate the associations between molecular genetics, electrophysiological, behavioral, and psychological characteristics of people. Keeping that in mind, we developed an internet resource including a database and tools for primary presentation of the collected data of genetic factors, the results of electroencephalography (EEG) tests, and psychological questionnaires. The sample of our study was 1010 people from different regions of Russia. We created the integrated ICBrainDB database that enables users to easily access, download, and further process information about individual behavioral characteristics and psychophysiological responses along with inherited trait data. The data obtained can be useful in training neural networks and in machine learning construction processes in Big Data analysis. We believe that the existence of such a resource will play an important role in the further search for associations of genetic factors and EEG markers of depression.

## 1. Introduction

Depression and anxiety are among the most common mental health disorders in a modern society. According to various estimates, they affect 10–15% of the world’s population. Depressive disorders significantly reduce the quality of life, damage health, and are even life-threatening due to an increased risk of suicide. In addition, depression also causes enormous economic harm. Despite the long history of elaboration of methods for diagnostic and treatment of depression, modern psychology and medicine face the challenge of identifying the genetic factors associated with this pathology [1]. Finding the genetic predisposition of psychological disorders is complicated due to the fact that the most of them are complex disorders associated with many various genes and environmental factors that influence dynamics of depression. In addition to genetic studies of the causes of depression, there has been an increasing accumulation of data on the neural basis of integrative brain activity and its pathology in depression, particularly by using of electroencephalography (EEG).

Consequently, during the last decade there has been a significant increase in studies of associations between genetic factors and brain activity indices derived from functional magnetic resonance imaging (fMRI) and EEG technologies. Many associations between genes and polymorphisms and brain activity features associated with susceptibility to psychopathology, including depression, have been found. However, most studies of both molecular-genetic and neurophysiological markers of depression and anxiety disorder have focused on identifying factors associated with the risk of disorders regardless of the region of residence. Such studies are carried out under the assumption that biological factors have the same influence on the risk of disorders for any climate and any social group of people. In this connection, most of the open databases, including EEG, genotyping data, and psychological surveys, do not take into account the regional specifics of the survey participants. However, there are also studies [2,3], according to which the associations between genetic and neurophysiological factors on the one hand, and the risk of mental disorders and psychological personality traits, on the other hand, are significantly modulated by climatic and ethnographic characteristics of people. The novelty of our database is that it contains the results of a survey of people from four regions (Siberian industrial city, rural areas of Western Siberia, the Republic of Tuva, and the Republic of Yakutia), which differ greatly both in climate and ethnic composition of the population. Using our data allows us to compare markers of depression and anxiety disorder in people living in a relatively temperate climate (Western Siberia), an extremely arid climate (Republic of Tuva), and an extremely cold polar climate (Yakutia). In addition, our results allow us to compare the differences between urban (Novosibirsk, Yakutsk) and rural populations of different regions, as well as between Turkic-speaking (Tatars, Yakuts, Tuvinians), Mongolian (Evenki) and Slavic (Russians, Ukrainians) ethnic groups. It is with this in mind that we have developed an internet resource including a database and tools for primary presentation of the collected data. Currently, our database contains EEG data, psychological questionnaires, and polymorphisms for more than 1000 subjects from different groups of Russian people by the region of residence. The database REST API (Representational State Transfer Application Programming Interface) can be accessed at: https://icbraindb.cytogen.ru/api-v2/.

In the following paper, we describe the current data available in the database, the methods by which they were obtained, and the database architecture. We also describe the database entities available for accession.

## 2. Materials and Methods

### 2.1. Subject Groups

The current version of the database stores information from a study conducted on 1010 subjects aged 13 to 82 years old (403 females, 407 males, and no gender specified for 200 subjects. The mean age was 26.4 ± 10.8; group mean ± standard deviation of mean). The subjects were divided into six groups, differentiated by the region of residence, ethnicity, and presence or absence of a diagnosed depressive spectrum disorder. These groups included (1) residents of Novosibirsk city and (2) residents of Bolotnoe (the rural area of Novosibirsk Region, a total of 420 people in two groups, 25.0 ± 9.2, 235 females, and 185 males), predominantly Caucasians of mixed ethnicity; (3) permanent residents of Yakutsk (main city of Sakha Republic, 50 people, 19.5 ± 3.7, 30 females, and 20 males), Yakuts, Evenks, and Yukaghirs in nationality; (4) permanent residents of the Khandyga settlement, located in the Arctic region of the Sakha Republic (Yakutia) (50 people, 24.0 ± 7.2, 26 females, and 24 males), Evenks and Yakuts in nationality; (5) migrants of different nationalities living in the Republic of Yakutia (50 people, 21.2 ± 3.1, all males); and (6) permanent residents of the Tuva Republic (122 people, 21.7 ± 3.0, 53 females, and 69 males), all Tuvinians in nationality.

We should separately mention a group of psychiatric patients diagnosed with major depression disorder (MDD, 61 persons, 42.8 ± 12.5, 25 females, and 36 males). Patients with an acute MDD episode were recruited from the inpatient and outpatient clinical departments of the Institute of Neuroscience and Medicine hospital. The mental health of this group was initially assessed using an unstructured interview with a psychiatrist according to the ICD-10 criteria [4], and later the severity of depression in patients was additionally assessed using the Structured Clinical Interview for DSM-IV [5] and DSM-V [6]. Exclusion criteria for this group were major medical illness, history of seizures, pregnancy, a history of substance abuse or dependence, as well as all contraindications against MRI. Specific psychiatric exclusion criteria for patients consisted of atypical forms of depression and any additional psychiatric disorder. MDD patients additionally completed the Hamilton Depression Rating Scale [7], and all participants filled in the Beck Depression Inventory (BDI-II) [8] and the trait anxiety scale from the State Trait Anxiety Inventory [9].

All subjects gave written informed consent to participate in the study in accordance with the ethical requirements of the World Medical Association’s Declaration of Helsinki. The protocol of the study was approved by the local ethical committee of the Institute of Neuroscience and Medicine (protocol No. 11 date of approval 20 October 2016). All subjects were renamed in the database by population-associated identifiers in order to maintain the confidentiality of personal data.

### 2.2. Genetic Data

Genetic material in the form of blood or buccal epithelium was taken from most subjects. Genomic DNA was extracted from blood samples with a QIAamp DNA Blood mini kit (Moscow, Russia) and PureLink PRO 96 (Waltham, MA, USA) from buccal epithelium. In total, genomic DNA was extracted from blood samples and buccal epithelium from more than 1200 individuals. A total of 960 DNA samples meeting the quality and quantity criteria for use in NGS sequencing was selected for further creation of targeted NGS libraries. Targeted sequencing of 164 genes was performed basing on this material. The genes were selected as a part of the gene network reconstruction and analysis according to the following factors: the presence of effects on neurological disease and the difference in expression in various ethnic groups shown in the literature [3]. Targeted sequencing libraries were prepared for these 164 genes and a new generation of high coverage sequencing was carried out on 820 subjects from six subject groups. The results of an allelic variant search on these data were transferred to our database.

### 2.3. EEG Data

For all subjects, EEG electrodes were placed on the head according to the international scheme 5–10%. In adult participants in Novosibirsk, EEG was recorded through 128 channels, in children through 64 channels. For participants from Tuva and Yakutia, EEG was recorded through 64 channels. The Cz channel was always used as a reference electrode, and the AFpz channel was used as a ground electrode. The resistance under the electrode was measured before the start of EEG recording and was less than 5 kΩ. The signals were amplified using Brain Products actiChamp amplifier (Gilching, Germany), with 0.1–100 Hz analog band pass and continuously digitized at 1000 Hz.

In the initial EEG recordings, the signal length depended on the experimental paradigm. For resting-state, EEG activity was recorded for 12 min (three times for 2 min with eyes open and 2 min with eyes closed) at a sampling frequency of 1000 Hz. Thus, the signal for each EEG channel had a length of 720,000 measurements for each participant. For other experimental conditions, the number of measurements depended both on the type of task and on the individual speed of its completion. The execution time for different tasks varied from 5 min (GO-noGo task) to 30 min (speech recognition task). In addition, for each person, part of the EEG recording fragments was excluded from the analysis due to unremovable artifacts.

Tests running during the EEG recording were computer programs that presented the examinee with stimuli (text, image, and sound) and recorded the examinee’s response of pressing the keyboard or joystick. The test experiment proceeded as follows: the subject with the EEG cap on was placed in a soundproof room with the lights turned off. The subject was instructed in advance about the rules of the experiment (what kind of stimuli the program would present and what kind of response the program would expect). Depending on the test program, instructions were given verbally by the experimenter or in writing on the computer screen. During such an experiment, an EEG was recorded from the subject simultaneously with task performance. The program also marked the EEG recording by marking the events of the experiment on it (e.g., presentation of a certain stimulus, beginning/end of the experiment, etc.). In addition, the program accumulated a protocol of the experiment, a table of the characteristics of the events of the experiment. For each program, a different set of characteristic columns was defined in the database. For example, all protocols contained a column under the response time (time interval from the presentation of stimulus to the moment of response), and in many programs with the presentation of the text information, protocols contained a column under the text that was displayed); the protocol was essentially a text representation of the experiment. When processing protocols, both individual (for one examinee) and group features of the experiment (for example, average response time or percentage of accuracy of answers (if the test contained “correct” and “incorrect” answers) was evaluated. After the EEG was recorded, the subjects underwent a set of psychological questionnaires for estimation of their personality traits, states under EEG recording, or predisposition to depressive disorder.

### 2.4. Questionnaires

The Big Five Factor Markers [10] were used to measure personality in a five-factor model. We also used the Eysenck Personality Profiler [11], which measures nine personality traits including anxiety and depression. In addition, the well-known State Trait Anxiety Inventory [9,12] and the Aggression Scale [13] were used. Emotional intelligence was assessed using the Barchard questionnaire validated by Knyazev et al. [14]. Two questionnaires were used to assess the expression of individualistic and collectivistic tendencies. The first one is the well-known Singelis Self Construal Scale [15] collective and independent self-concept. The second questionnaire measured affective tendencies selectively towards immediate family members or a loved one [16] (RISC, The Relational-Interdependent Self-Construal). The Beck Depression Inventory [8], the Adult Behavior Checklist [17], and the WHO Self-Reporting Questionnaire (SRQ20) [18] were used to assess the severity of depressive symptoms.

The list of tests in the database included behavioral tests for aggression, anxiety, depression, and impulsiveness, where one had to press buttons left and right depending on the word combination shown on the screen; Hunt/Zoo test, Moral test, Socgame test, Gend, Emot, P300, and language test for grammatical errors. A detailed description of the tests and the items they produced are given in the ‘testWithEGG’ table.

## 3. Results

### 3.1. Database Content

The database provides a comprehensive access to depersonalized data on each individual, including genetic traits, fMRI, and psychological questionnaire summaries data via REST API. Our choice to focus on REST API implementation instead of classical graphical user interface implementation is primarily motivated by a desire to give users a free choice to construct search requests whether they are simple or complex without limitations of standard use-cases that are typical for the latter. This approach allows the use of tools for data analysis [19], including the tools for building summary charts, which have recently become more and more widely used.

EEG data-*.set and/or *.fdt files. Background recording data are presented for all subjects, with the subject sitting with their eyes open for a long time and then sitting with their eyes closed for a long time. There are event markers on the EEG data.

Resting-state EEG was recorded in every participant without any external functional load for 12 min (three two-minute intervals with eyes closed and three two-minute intervals with eyes opened). Not only can the resting-state EEG data be found in the database but also the EEG data taken during the tests listed below. These EEG experiments were not performed on all subjects but only on some of the participants. The tests during the EEG recording were in the form of computer games. One example of such a test is the subject undergoing examination was sitting in front of a computer screen on which various signals were displayed, to which they should react by pressing buttons, depending on the situation. There are Resting State, Motor-Imagery, Emotion-Recognition, Event Related Potentials [ERPs], and Eye-blinks/movements EEG data in the ICBrainDB.

In addition to the tests in the EEG recording, our database stored the results of processing the psychological questionnaires that the same subjects took before and after the EEG recording. The list of questionnaires is stored in the table test, with the results of processing of questionnaires in the table Test Summary.

The allelic variant information stored in our database includes the localization on the chromosome, the variant locus reference, the mutated allelic variant, and the genotype (homozygote or heterozygote) for each subject. Currently, 6316 allelic variants are stored in the database.

### 3.2. Database Structure and Access

The database has a relational structure and is implemented with PostgreSQL. It consists of nine tables describing the main elements of the database (Figure 1).

The database table structure reflects the properties and relationships of the following subject area objects:human-subject contains the key parameters of a subject.mutation-data on mutations in a particular subject detected in a gene located on a particular chromosome; ref_nucl parameter indicating reference locus variant; and type parameter indicating genotype (homozygote or heterozygote).Test-a questionnaire contains information about the name of the questionnaire and a description of how it was used; alias–is the common name of the questionnaire.testQuestion, a table that contains questions associated with the questionnaires, with the order of the questions.testResponseType, a table that describes the answer choices allowed in the questionnaire. There can be either a value between 1 and 5, or some ranking [“yes”, “probably yes”, “probably no”, “no”, and “definitely no”].testSummary contains the results of the examiner’s processing of the examinee’s questionnaire responses; alias–is the common name of the summary. With a single questionnaire there could be more than one testSummary. These connections are represented in the ‘testSummary2test’ table.testResults contains the results of responses to specific questions associated with valid response options.EEG_file-table lists paths to EEG data files for a specific examinee. It is used for creating a download URL link.

### 3.3. Data Access

A REST API was implemented to access the data (icbraindb.cytogen.ru/api-v2/). To access this interface, one can use both programming languages (like Python (Beaverton, USA)) and an engineering modelling environment (Matlab (Natick, MA, USA) or Octave (https://octave.org/doc/v6.4.0/), for example), which have appropriate libraries. The result of the API request is a JSON object, which can be parsed by standard tools. In this way, one can customize their own scripts (A data access example is given in Appendix A) to analyze the data and generate summary images (Figure 2).

A description of the key API queries is provided below:

SUBJECTS:/api-v2/human: list of all subjects./api-v2/human/<string:id>: information about a particular subject (here and after the ‘<string:id>’ part of the path means the identifier of the entity)./api-v2/human/<string:id>/mutations: information about this subject’s mutations./api-v2/human/<string:id>/files: information about the subject’s available EEG files.
GENES/api-v2/genes: list of genes in the database./api-v2/genes/<string:id>: information on a particular gene, contains a list of mutations of this gene in the database.QUESTIONS/api-v2/questionnaires/: list of questionnaires./api-v2/questionnaires/<string:id>: selected questionnaire information, which includes the list of questions.QUESTIONNAIRE RESULTS/api-v2/summaries/: list of available subject questionnaire summaries./api-v2/summaries/<string:id>: list of selected questionnaire summary values for all subjects./api-v2/questionnaire-results/: list of all questionnaire summary values in database./api-v2/questionnaire-results/<string:id>: selected questionnaire summary value.MUTATIONS/api-v2/mutations: the list of mutations in the database./api-v2/mutations/<int:id>: information on a particular mutation.FILES/api-v2/files: list of EEG files available in the database./files/<string:id>: link to download a specific EEG file.

## 4. Discussion

Our database contains a large amount of integrated data on genetic factors and EEG markers of emotional states from more than 1010 subjects.

Due to the uniqueness of our approach and the amount of data stored, it is challenging to find any databases with similar characteristics to ICBrainDB. Most multilevel databases with more than 500 subjects, such as MK4MDD [20] or PCG [21], do not provide the raw experimental data but only the results of their processing–found associations and relationships, which makes it difficult to analyze the data with different methods. Databases that store multiple EEG datasets from different studies and provide the raw data, such as Zenodo (https://zenodo.org, accessed on 1 November 2021), Dryad (https://datadryad.org), OpenNeuro (https://openneuro.org) or Donders Repository (https://data.donders.ru.nl), do not store any data other than EEG files. Most of the datasets also could have been made with different recording methods, which would make it difficult to analyze data from different datasets. Meanwhile, ICBrainDB also provides multilevel information about the subject, from individual allelic variants and behavioral data from psychological questionnaires to results from different EEG tests. The overwhelming majority of independent datasets do not store data from more than 100 subjects.

Geneticists may be interested in data on genetic variations in people from different regions of residence and ethnicities, such as single nucleotide polymorphisms and indels. Numerous studies have shown the involvement of allelic variants in human health, particularly in diseases of nervous systems [22,23,24,25]. Allelic variants derived from sequencing data combined with physiological EEG data allow direct testing of all genetic variants presented in a sample set for association with a given disease or trait.

Psychologists may be interested in a large volume of raw and preprocessed EEG data. It is possible to search for associations between EEG markers of depression and other human emotional states and the effects of genetic factors on them. Our database contains the results of a variety of tests aimed at studying various human emotional states. In particular, the EEG data obtained from the stop-signal test allows us to test the relationship between the activation and inhibition processes in the brain, which ensure the voluntary control of behavior.

The ICBrainDB is a growing database of physiological, psychological, and genetic data associated with emotional perception and depression. It has already stored about 2.4 TByte of EEG data from 875 subjects and processed target sequencing data from these subjects. The integrated database will enable users to easily access, download, and further process information about individual behavioral characteristics and psychophysiological responses along with inherited trait data. We believe that the existence of such a resource will play an important role in the further search for associations of genetic factors and EEG traits on depression. Furthermore, we are planning to maintain the ICBrainDB by adding both new subjects and raw EEG data processing results to provide users with as much research information as possible.

## Figures and Tables

**Figure 1 jpm-12-00053-f001:**
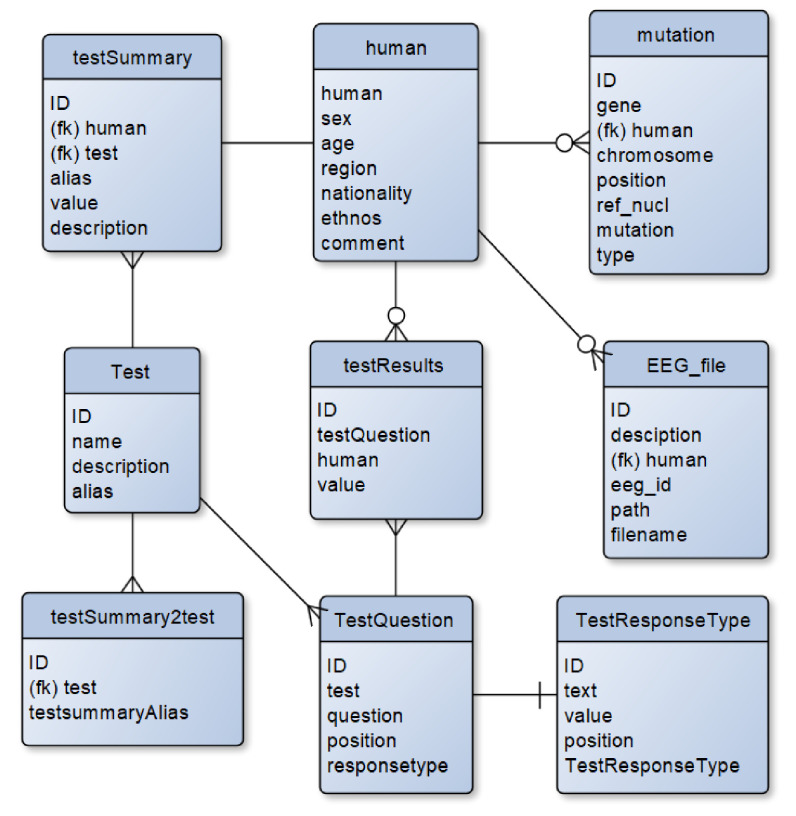
The database structure in “crow’s feet” notation. The characters “(fk)” in the table fields indicate the fields that are foreign keys.

**Figure 2 jpm-12-00053-f002:**
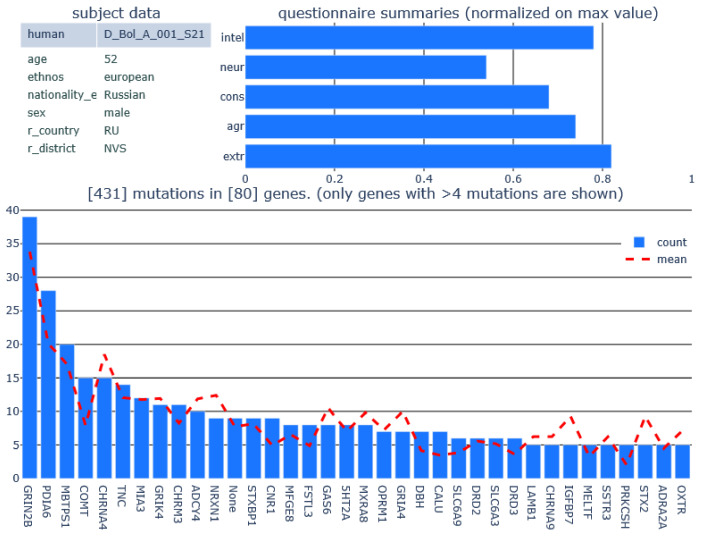
Patient data summary built with python script (see the Appendix A) on ICBrainDB REST API. Basic data (left upper corner). Selected test summaries data from Goldberg’s Big Five Factor Markers normalized on the largest value of each summary in the database (right upper corner). The plot at the bottom with the patient genes that have mutation (on X axes) vs the amount of these genes mutations (bars, Y axes) and mean mutations amount of the gene in a database (red dashed curve).

## Data Availability

The data presented in this study are openly available in https://icbraindb.cytogen.ru/api-v2/.

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
