# Peer review of "ICBrainDB: An Integrated Database for Finding Associations between Genetic Factors and EEG Markers of Depressive Disorders"

_jpm, 2022, doi:10.3390/jpm12010053_

Round 1

Reviewer 1 Report

  1. Statistical analysis is missing in this study. Did you calculate sample size among different subjects?
  2. Discussion is not adequate. Better to compare the proposed work with available depression EEG datasets or provide brief disccussion. Have you seen following studies for your literature work? A novel computer-aided diagnosis framework for EEG-based identification of neural diseases, Depression recognition based on the reconstruction of phase space of EEG signals and geometrical features
  3. Please arrange dataset in .mat format and arrange dataset in better and clear way for experimentationl purpose.
  4. Better to explain the signal length and channels information for new users
  5. English language need improvements.

Author Response

We thank the reviewer for their recommendations and comments. We accept the most of the reviewer’s suggestions and have changed the manuscript according to them.

  1. Statistical analysis is missing in this study. Did you calculate sample size among different subjects?

 Response 1. Our study indeed misses the statistical analysis since the presented work did not imply the presentation of a hypothesis and therefore we did not conduct a statistical analysis of the selection. We just represent the database, not the results of its processing.

  1. Discussion is not adequate. Better to compare the proposed work with available depression EEG datasets or provide brief disccussion. Have you seen following studies for your literature work? A novel computer-aided diagnosis framework for EEG-based identification of neural diseases, Depression recognition based on the reconstruction of phase space of EEG signals and geometrical features

Response 2. Due to the novelty of our approach it was challenging to find some similar databases to compare, but we have added the comparison based on data type availability.

  1. Please arrange dataset in .mat format and arrange dataset in better and clear way for experimentationl purpose.

Response 3. Due to the of size of our dataset – around 2,5 Tb, we are unable to rearrange the whole dataset at the moment. We suggest to use the script in the Supplementary to change the files format to .mat. Later we will add new data in .mat format.

  1. Better to explain the signal length and channels information for new users

Response 4. We changed the description of the EEG hopefully better explaining the signal length and channels information.

  1. English language need improvements.

Response 5. The final version of the manuscript was edited by a native English speaker.

Reviewer 2 Report

The paper's topic is exciting, and creating a database available to researchers could help define methodologies for studying this pathology. However, the article in the current version cannot be accepted. It is necessary that:
1. the authors improve the introduction by better-describing the state of the art. They also have to highlight if there are other similar DBS. In the introduction, it would be preferable to describe the structure of the paper.
2. Image quality needs improvement.
3. The novelty of the work must be highlighted and discussed in a better way.
4. The given link HTTP: // ic-288 braindb.cytogen.ru/api-v2/ does not work.

Author Response

We thank the reviewer for their recommendations and comments. We accept the most of the reviewer’s suggestions and have changed the manuscript according to them.

  1. the authors improve the introduction by better-describing the state of the art. They also have to highlight if there are other similar DBS. In the introduction, it would be preferable to describe the structure of the paper.

Response 1. We have changed the introduction, describing in more detail the significance of the work and adding the structure of the article in the introduction. We have also added comparison with other databases in the discussion. 

  1. Image quality needs improvement.

Response 2. We have replaced the images with better quality.

  1. The novelty of the work must be highlighted and discussed in a better way.

Response 3. We have rewritten the discussion and introduction to better highlight the novelty of our work.

  1. The given link HTTP: // ic-288 braindb.cytogen.ru/api-v2/ does not work.

Response 4. The specified link does not work most likely due to a text error in the first version manuscript file. The right link is –

https://icbraindb.cytogen.ru/api-v2/

Round 2

Reviewer 1 Report

I am not satisfied with the revisions and cannot trust on the dataset without proper statistical analysis.

Statistical analysis is missing in this study. Did you calculate sample size among different subjects? Have a look on following authors study to learn about statistical analysis. Also provide discussion of your study in comparison with the suggested study

 “A novel computer-aided diagnosis framework for EEG-based identification of neural diseases, Depression recognition based on the reconstruction of phase space of EEG signals and geometrical features”

Author Response

Point 1 I am not satisfied with the revisions and cannot trust on the dataset without proper statistical analysis.

Statistical analysis is missing in this study. Did you calculate sample size among different subjects? Have a look on following authors study to learn about statistical analysis. Also provide discussion of your study in comparison with the suggested study “A novel computer-aided diagnosis framework for EEG-based identification of neural diseases, Depression recognition based on the reconstruction of phase space of EEG signals and geometrical features”

Response: We thank the Reviewer 1 for their recommendations and comments. However, we should mention that we did not conduct any analysis in this manuscript and only provide the description of the database. The suggested statistical analysis – sample size count, depends on both statistic distribution of parameter and its dispersion for receiving the statistically significant results. Many different indexes (psychological, genetic and neurophysiological) have been putted in our data base. Sample size should be calculated individually for each of these parameters.

The type of analysis the Reviewer 1 suggested (requiring estimating of several hundred additional indexes) deserves a separate consideration in a separate study, which we would be happy to conduct in future.

Yet we have found that EEG [1–3] and medical-genetics data repositories [4] as well as a number of papers published in high IF journals [5–8] also do not provide statistical estimations of the above-mentioned type. Considering the above we only have described the number of the participants in the present study.

Regarding the comparison with the recommended articles “A novel computer-aided diagnosis framework for EEG-based identification of neural diseases” and “Depression recognition based on the reconstruction of phase space of EEG signals and geometrical features” – we do think that our paper’s scope is different from theirs. Aforementioned papers describe the novel methods of depression recognition based on EEG characteristics, which can provide a very perspective approach to the further analysis of our dataset. But within the context of the present paper, we believe that the comparison would not provide any meaningful addition to the manuscript.

  1. Markiewicz, C.J.; Gorgolewski, K.J.; Feingold, F.; Blair, R.; Halchenko, Y.O.; Miller, E.; Hardcastle, N.; Wexler, J.; Esteban, O.; Goncavles, M.; et al. The OpenNeuro resource for sharing of neuroscience data. Elife 2021, 10, doi:10.7554/eLife.71774.
  2. Aguilar, M.; Congedo, M.; Minguez, J. A data-driven process for the development of an eyes-closed EEG normative database. In Proceedings of the 2011 Annual International Conference of the IEEE Engineering in Medicine and Biology Society; IEEE, 2011; pp. 7306–7309.
  3. Katsigiannis, S.; Ramzan, N. DREAMER: A Database for Emotion Recognition Through EEG and ECG Signals From Wireless Low-cost Off-the-Shelf Devices. IEEE J. Biomed. Heal. Informatics 2018, 22, 98–107, doi:10.1109/JBHI.2017.2688239.
  4. Huser, V.; Cimino, J.J. Linking ClinicalTrials.gov and PubMed to Track Results of Interventional Human Clinical Trials. PLoS One 2013, 8, e68409, doi:10.1371/journal.pone.0068409.
  5. Zhdanov, A.; Atluri, S.; Wong, W.; Vaghei, Y.; Daskalakis, Z.J.; Blumberger, D.M.; Frey, B.N.; Giacobbe, P.; Lam, R.W.; Milev, R.; et al. Use of Machine Learning for Predicting Escitalopram Treatment Outcome From Electroencephalography Recordings in Adult Patients With Depression. JAMA Netw. Open 2020, 3, e1918377, doi:10.1001/jamanetworkopen.2019.18377.
  6. Kennedy, S.H.; Lam, R.W.; Rotzinger, S.; Milev, R. V.; Blier, P.; Downar, J.; Evans, K.R.; Farzan, F.; Foster, J.A.; Frey, B.N.; et al. Symptomatic and Functional Outcomes and Early Prediction of Response to Escitalopram Monotherapy and Sequential Adjunctive Aripiprazole Therapy in Patients With Major Depressive Disorder. J. Clin. Psychiatry 2019, 80, doi:10.4088/JCP.18m12202.
  7. Prakash, B.; Baboo, G.K.; Baths, V. A Novel Approach to Learning Models on EEG Data Using Graph Theory Features—A Comparative Study. Big Data Cogn. Comput. 2021, 5, 39, doi:10.3390/bdcc5030039.
  8. Cai, H.; Qu, Z.; Li, Z.; Zhang, Y.; Hu, X.; Hu, B. Feature-level fusion approaches based on multimodal EEG data for depression recognition. Inf. Fusion 2020, 59, 127–138, doi:10.1016/j.inffus.2020.01.008.

Reviewer 2 Report

The authors accepted most of the reviewer's suggestions.

Author Response

We thank the Reviewer 2 for their positive decision.

Round 3

Reviewer 1 Report

I have no further questions.